# Association of expenditure on ultra-processed foods and beverages and anthropometric indicators in Mexican children: A longitudinal study

**Mauricio Hernández-F[1], Sonia Hernández-Cordero[1], Mishel Unar-Munguia[2],
Wilfrido A. Gómez-Arias[3], Erika Lozano-Hidalgo[1], Lidia Sarahi Peña-Ruiz[1],
Graciela Teruel-Belismelis[4]*¤**

1 Research Institute for Equitable Development, Universidad Iberoamericana, Ciudad de México, México,
2 Center for Nutrition and Health Research, National Institute of Public Health, Cuernavaca, Morelos,
México, 3 Institute of Applied Research and Technology, Universidad Iberoamericana, Ciudad de México,
México, 4 Social Sciences Division, Universidad Iberoamericana, Ciudad de México, México

¤Current Address: División de Estudios Sociales, Universidad Iberoamericana, Ciudad de México, México.
* chele.teruel@ibero.mx

org/10.1371/journal.pone.0317831

University, CHINA

**Peer Review History:** PLOS recognizes the
benefits of transparency in the peer review
process; therefore, we enable the publication
of all of the content of peer review and
author responses alongside final, published
articles. The editorial history of this article is
available here: https://doi.org/10.1371/journal.
pone.0317831

## Abstract

The prevalence of obesity in Mexico has been rising dramatically from school age onward.
The high consumption of ultra-processed food has been identified as a contributing factor.
We explored the longitudinal association between household expenditure on ultra-processed foods and beverages (UPF) and changes in anthropometric indicators of obesity among Mexican children aged 5 to 10 years in 2002. We used data from the Mexican Family Life Survey (MxFLS), a longitudinal, probabilistic, multipurpose, and representative survey of the Mexican population conducted in 2002, which reports household expenditure on the main food and beverage groups, as well as anthropometric indicators and sociodemographic characteristics of household members, across three rounds surveyed between 2002 and 2012 (n = 2,677). The exposure variable was UPF expenditure, categorized into tertiles, and the outcomes studied were BMI z-score for age, waist circumference, and waist-to-height ratio. We estimated random effects models and generalized estimating equation models for longitudinal data. Using an interaction term between tertiles of UPF expenditure and survey rounds, we found that household membership in the middle and upper tertiles of UPF expenditure in 2002 was associated with an increase in waist circumference and waist-to-height ratio, particularly after three years of follow-up. For instance, the middle tertile of UPF expenditure was associated with an increase of 4.43 centimeters in waist circumference compared to the low tertile of UPF expenditure after three years of follow-up (p < 0.01). Our findings suggest that higher UPF expenditure in households with children aged 5–10 years drives abdominal obesity in the short and medium term, underscoring the need for comprehensive policies to limit the purchase and consumption of UPF from an early age.

**Data availability statement:** The data under-lying the results presented in the study are available from INEGI (https://www.inegi.org.mx/app/indicesdeprecios/Estructura.aspx?idEstructura=112000200070&T=%EF%BF%BD-ndices+de+Precios+al+Consumidor&ST=IN-PC+Nacional). Data retrieved from MxFLS has been provided in the form of Supporting Information files.

**Funding:** MHF received a grant (ID 0061) from the Research Institute for Equitable Development (EQUIDE for its Spanish acronym) at Universidad Iberoamericana (https://equide.org/), within the framework of its internal Call for Research Project Funding 2022. The funders had no role in study design, data collection and analysis, decision to publish, or preparation of the manuscript.

**Competing interests:** The authors have declared that no competing interests exist.

## Introduction

In recent decades, chronic noncommunicable diseases such as diabetes, ischemic disease, and chronic kidney disease have borne most of the global burden of disease [1]. Several diet-related risks have been identified as drivers of this transition [2].

One of the proposals that have been made to characterize dietary patterns associated with health and nutrition outcomes is the NOVA classification, which categorizes foods into four groups according to their degree of processing [3]. Group 4, defined as ultra-processed foods and beverages (UPF), includes products created by industrial processing, from formulations based on processed substances or synthesized from other organic sources that provide energy, sugars and/or saturated fats at low cost, whose consumption should preferably be avoided to reduce dietary risks [4].

Several studies in adults have found a correlation between UPF consumption and obesity and metabolic alterations; this evidence been generated primarily from longitudinal studies [5–8]. On the other hand, a crossover randomized controlled experiment, which offered ultra-processed and non-ultra-processed food options to 20 people for 14 days each, on demand, found that the diet abundant in UPF caused a markedly higher caloric intake and a weight difference of 1.8 kg [9]. The proposed mechanisms highlight the energy density that leads to high caloric intake with relatively low satiety signaling, the speed at which their calories can be consumed, and the influence of their components on the reward system [10]. However, in children and adolescents, the evidence is relatively scarce and inconclusive.

A study based on a cohort of children in Brazil found a positive association between the contribution of UPF to energy intake and the change in waist circumference between pre-school and school age [11]. In contrast, a study on adolescents in Brazil found no significant difference in the expected trajectories of body mass index (BMI) between the first and fourth quartiles of UPF consumption [12].

In Mexico, the prevalence of childhood weight excess and obesity increased markedly between 1999 and 2012 [13]. UPF expenditure in Mexico has also experienced a significant increase over the years. In 1984 it represented an average of 10.5% of dietary energy intake while in 2006 it increased to 22.3% of total kcal. Household purchases experienced slight variations in subsequent years, reaching 23.7% of total caloric intake in 2012 and dropping to 23.1% in 2016 [14]. The association between indicators of UPF consumption and indicators of obesity needs to be studied because children and adolescents are at a stage of growth and formation of key habits for the prevention or development of diet-related chronic diseases in the future [15].

The study aims to leverage data from a longitudinal survey, which includes anthropometric indicators over time, to examine the association between UPF expenditure and obesity in children. This analysis estimates the longitudinal association of daily household expenditure in UPF per equivalent adult with anthropometric indicators such as 1) BMI z-score for age, 2) waist circumference, 3) waist circumference to height ratio (referred to subsequently as waist-to-height ratio), in children aged 5 to 10 years in 2002 in a representative sample of the Mexican population.

## Materials and methods

### Study sample

This study is a longitudinal analysis based on data from the Mexican Family Life Survey (MxFLS), which is a population-based, multipurpose survey, representative of the Mexican population in 2002 at national, urban/rural, and regional levels. The survey has a probabilistic, multistage, stratified, clustered, multistage design. Its detailed description can be found

elsewhere [16–18]. The baseline MxFLS was conducted in 2002 (MxFLS-1) and included 8440 households and 35677 individuals (questionnaires were administered to all household members). The second round was conducted between 2005 and 2006 (MxFLS-2), and the third round was conducted between 2009 and 2012 (MxFLS-3). Re-contact rates at the household level were approximately 90% [16,17]. The longitudinal design of the MxFLS makes it possible to monitor the development of individuals for up to a 10-year period [19].

For this study, we used data obtained from a sample of children aged 5 to 10 years 11 months in the MxFLS who had complete information in the first round (2002). This age group is of interest because precisely during their school years, children experience a notable increase in obesity in Mexico [13], and the food environment in Mexico, especially the school environment, has been described as obesogenic [20]. As the intention to maintain or lose body weight has been associated with changes in eating behaviors [21], we excluded those cases that had an anthropometric measurement indicative of obesity in the first round of the MxFLS. We also excluded those children with extreme values of birth weight (<2.5 or >4 kg) because it is an important predictor of metabolic disturbances and nutritional status in the life cycle [22]. We also excluded children with health described as poor or very poor, with 15 or more days without usual activities due to illness, who had a serious accident in the month prior to the interview, or with an unreliable BMI indicator according to World Health Organization (WHO) criteria [23]. The final sample size was 2,677 children, of whom 1,474 were followed up in the second round and 1,502 in the third round of the MxFLS (Fig 1).

## Exposure variable

The MxFLS collected information on weekly household expenditure on the 37 most consumed foods or food groups [24], and we stratified them according to the NOVA classification to identify the UPF [4]. Fig 2 describes the foods included in each group.

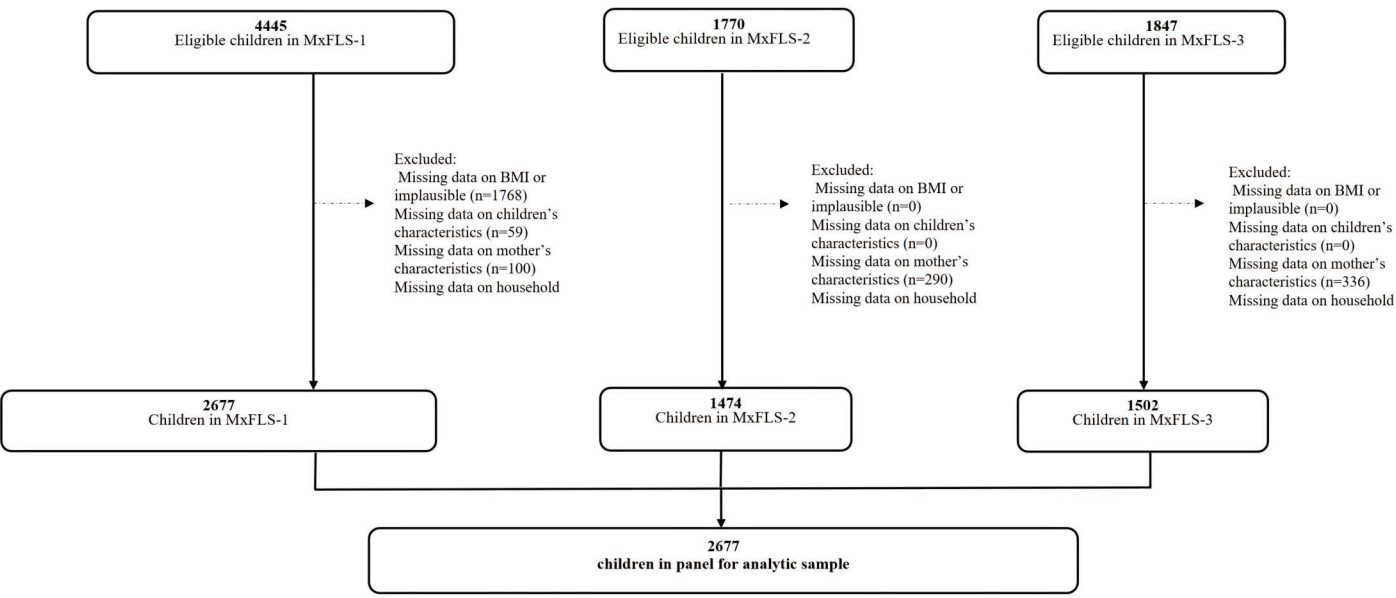

**Fig 1. Flowchart of the study sample.** MxFLS-1 the Mexican Family Life Survey wave 1 (2002); MxFLS-2 the Mexican Family Life Survey wave 2 (2005–2006); MxFLS-3 the Mexican Family Life Survey wave 3 (2009–2012). Children's characteristics: BMI, age, sex. Mothers' characteristics: BMI, marital status, education. Household characteristics: wealth index, size of locality, urbanity, household expenditure, size of household.

It was not possible to classify expenditure on food and beverages consumed outside the home (breakfast, lunch, and dinner) reported in the MxFLS according to the NOVA classification. However, expenditure on food outside the home was taken into account to determine the total expenditure on the purchase of food other than UPF (although it is not possible to classify it, the highest percentage is considered to be on non-UPF). We also excluded two food groups reported in the MxFLS, because it is not clear which would be the most appropriate NOVA group, given that they include products from several NOVA groups: (a) processed bread loaves, flour, flakes, corn grain, tortilla dough; (b) juices, purified water, beer, tequila, rum, and powdered drink mixes.

We estimate daily spending per adult equivalent on UPF products (group 4) at constant March 2002 prices, in Mexican pesos. The expenditure reported in the second and third rounds of the MxFLS was adjusted to constant 2002 prices using the National Consumer Price Index reported by the National Institute of Statistics and Geography (INEGI for its Spanish acronym) [25]. The adjustment per equivalent adult was made according to the estimates of the Engel curves made by Teruel, Ruvalcaba, and Santana with data from the National Survey of Household Income and Expenditures [26]. Finally, the exposure variable was determined as the tertiles of the expenditure in UPF in the MxFLS-1.

## Outcome variables

Anthropometric measurements of weight, height, and waist circumference were obtained by standardized personnel in the three rounds of the MxFLS. Weight was measured with a digital scale (Tanita) to the nearest 0.1 kg and height was measured to the nearest 0.1 cm with a SECA stadiometer. We considered three outcome variables.

**BMI z-score for age.** This variable was calculated using the WHO AnthroPlus Software, which makes it possible to assess the growth of children and adolescents in a way that makes them comparable to each other [27]. It has been suggested that BMI z-scores for age may be inadequate to analyze growth over time, so we also analyzed BMI as a sensitivity analysis [28,29].

**Waist circumference (WC).** This was obtained directly from anthropometric measurements and reported in centimeters.

| Group 1 | Group 2 | Group 3 | Group 4 |
|---|---|---|---|
| onion, potato, chili, banana, apple, oranges, tangerine, lemon, grapefruit, peach, lettuce, carrot, pumpkin, avocado, nopal, pasta soup, rice, chickpeas, lentils, lima beans, beef, pork, fish, seafood, coffee, corn tortillas, chicken, milk, eggs, tomato, beans | vegetable oil and white sugar | canned tuna or sardines, cheeses, or bun | cookies, powdered milk, butter, cream, lard, ham, sausage, pastries, candy, chips and soft drinks. |

**Fig 2. Classification of foods and beverages with reported expenditures in the MxFLS based on the NOVA classification[a].** [a]Two food groupings reported in the MxFLS were excluded because it is not clear which would be the most appropriate group for them: (1) bread loaf, flour, flakes, corn grain, masa; (2) juices, purified water, beer, tequila, rum, and powder for preparing water.

**Waist-to-height ratio.** This variable was estimated as the quotient between WC and height [30].

The WC and waist-to-height ratio have been used in other studies as indicators of abdominal obesity [31,32]. These practical tools have shown adequate discriminatory power for body fat in children and adolescents of both sexes. Waist circumference identifies abdominal fat distribution, while waist-to-height ratio identifies total and abdominal body fat, in addition to cardiovascular risk associated with obesity [33]. Both measurements are common tools in pediatric assessment [31].

## Covariates

We consider in the models some adjustment variables at the individual, household, and environmental levels that could be related to exposure and outcomes [22] to reduce the likelihood of obtaining a biased estimate of the association between UPF expenditure and anthropometric measurements.

**Characteristics of the child.** The age of the children was included in completed years and sex as a binary variable (0 = female, 1 = male). The proportion of time reported watching television was included as a proxy for sedentary lifestyle and was estimated from the total weekly hours spent among a set of nine types of activities [34].

**Characteristics of the mother.** Mother´s years of schooling was estimated as years of education completed (0 = no education; 1 = from 1 to 6 years; 2 = over 6 years). Mother´s BMI was obtained from the mother's weight and height and categorized (1 = normal or underweight; 2 = overweight; 3 = obese) using World Health Organization (WHO) cut-off points. Mother´s marital status was obtained through information provided by the mother and categorized as: married, single, widowed, and divorced/separated.

**Household characteristics.** The size of household (number of household members) was considered to take into account differences between large and small households. The proportion of total food expenditure allocated to the purchase of non-UPF food was included. A wealth index was calculated based on housing characteristics (floor material, roof material, number of rooms, availability of water inside the dwelling, use of gas) and possession of assets (own home, second home, vehicle, household appliances, etc.), based on the Principal Component Analysis method and following the methodology of Filmer and Pritchett [35]. The wealth index was categorized into quintiles. The size of the locality in which the household is located was included in four categories according to the number of inhabitants: 1 (≤ 2 499), 2 (2 500 – 14 999), 3 (15 000 – 99 999), and 4 (≥100 000).

Finally, indicator variables were included for the MxFLS rounds from which the information is derived. Since expenditure results from the quantity purchased and the price of food, we also took into account the relative price of fruits and vegetables compared to the price of UPF, based on the National Consumer Price Index [25].

## Statistical analysis

A descriptive analysis of the sample characteristics was performed, where continuous variables were expressed as means and standard deviation (SD), while categorical variables were expressed as percentages. It was also verified that there was no multicollinearity of the independent variables.

To estimate the association between UPF expenditure and anthropometric indicators, we estimated two types of models for each of the outcomes. We estimated random-effects linear regression models for longitudinal data [36] by applying the STATA xtreg command. We also estimated generalized estimating equation models [37] (hereafter referred to as GEE models)

using the xtgee command, assuming unstructured correlation, gamma family and link log (except for BMI z-score for age where the assumption is Gaussian family and link identity). These models correct correlations between repeated measures and loss to follow-up [38].

Baseline tertiles of UPF were maintained constant to analyze the association with changes in obesity indicators in MxFLS-2 and MxFLS-3 compared to MxFLS-1. Therefore, the main parameters of interest are the coefficients of the interactions between the tertiles of UPF expenditure in 2002 and the MxFLS rounds. The statistical significance levels considered are the usual 1%, 5%, and 10%. The hypothesis tests were two-tailed.

All analyses were estimated with Stata software version 13.0 (College Station, TX).

## Ethical aspects

This study was based on a secondary analysis of the MxFLS data; the original protocol has the approval of the Ethics and Research Committee of the Mexican National Institute of Public Health (INSP for its Spanish acronym) and the Mexican National Institute of Perinatology (INPer for its Spanish acronym). All participants signed a letter of informed consent and assent.

## Results

Table 1 presents the characteristics of the children in the sample and their households in the three rounds of the MxFLS. Of the 2,677 children aged 5–10 years in MxFLS-1, 55% could be re-contacted in MxFLS-2 and 56% in MxFLS-3. Of note, mean BMI z-score values for age, waist circumference, and waist-to-height ratio rebounded in round 2, although by round 3 they remained more stable or even decreased. There was also a gradual increase in the proportion of time reported for non-school activities spent on watching television. The proportion of mothers with primary schooling or less was around 50%, although it showed a reduction in round 3. There was also a notable increase in the prevalence of obesity in mothers, from 28.6% in the first round to 40.8% in the third round.

Table 2 shows the average daily UPF expenditure per equivalent adult, in Mexican pesos at constant 2002 prices, in each round of the MxFLS, classifying the sample according to the tertiles of expenditure in MxFLS-1. In relative terms, UPF spending is much higher in the upper tertile compared to the other two tertiles in 2002, but the gaps narrowed in rounds 2 and 3 of the MxFLS.

The two types of models estimated for each of the three outcomes studied are shown in Table 3. Of note, the interaction terms between round 2 and the middle tertile are significant at the 1% significance level for the outcomes of waist circumference and waist-to-height ratio, both in the random-effects model and in the GEE model. Also, the interactions of the high tertile with round 2 are statistically significant, although the coefficients are slightly smaller. As an example of the interpretation of the coefficients, the middle tertile of UPF spending is associated with 4.43 centimeters more waist circumference than the low tertile, after controlling for covariates, as well as for the differences associated with the tertiles at baseline and for the increase observed when moving from round 1 to round 2 associated with factors not considered in the model. This increase associated with UPF expenditure represents 0.38 standard deviations of the waist circumference measurement in MxFLS-2.

Regarding the BMI z-score for age, we found heterogeneous results. Only the interaction term between the high tertile and round 2 reaches statistical significance at a 1% significance level, while the interaction term between the middle tertile and round 3 reaches significance at 5% in the GEE model, although the coefficient is negative.

**Table 1. Characteristics of the study sample. MxFLSᵃ, Mexico.**

| | Round of MxFLS | | |
|---|---|---|---|
| | **MxFLS-1** | **MxFLS-2** | **MxFLS-3** |
| | **n = 2,677** | **n = 1,474** | **n = 1,502** |
| **Children's characteristics** | | | |
| BMI z-score for age | 0.38 ± 1.11 | 0.93 ± 1.3 | 0.52 ± 1.1 |
| Waist circumference (cm) | 56.0 ± 7.5 | 72.3 ± 12.0 | 77.7 ± 10.0 |
| Waist-to-height ratio | 0.45 ± 0.04 | 0.47 ± 0.07 | 0.48 ± 0.6 |
| Age (years) | 7.6 ± 1.6 | 9.1 ± 1.5 | 14.6 ± 1.7 |
| Proportion of time reported watching television | 36.6 ± 21.8 | 43.3 ± 24.4 | 46.5 ± 25.8 |
| Sex (female) (%) | 51.6 | 53 | 51.8 |
| **Mothers' characteristics** | | | |
| Mother's years of schooling | | | |
| No education (%) | 7.6 | 6.8 | 4.8 |
| 1–6 years (%) | 47.1 | 47.1 | 45 |
| >6 years (%) | 45.1 | 46 | 50 |
| Mother's BMI | | | |
| Underweight or normal weight (%) | 31.8 | 24.6 | 20.6 |
| Overweight (%) | 39.5 | 41.1 | 38.5 |
| Obesity (%) | 28.6 | 34.1 | 40.8 |
| Mother's marital status | | | |
| Married/cohabiting (%) | 91 | 88.1 | 85.9 |
| Divorced/separated (%) | 4.7 | 6.3 | 8.2 |
| Widow (%) | 1.3 | 1.4 | 2.8 |
| Single (%) | 2.8 | 4.1 | 3 |
| **Household characteristics** | | | |
| Size of household (number of members) | 5.7 ± 1.9 | 5.8 ± 1.9 | 5.8 ± 2.1 |
| Proportion of food expenditure on non-ultraprocessed food and beverages | 83.3 ± 11.7 | 86.6 ± 9.5 | 88.0 ± 8.1 |
| Wealth Index (quintiles) | | | |
| I | 20.5 | 20.5 | 20 |
| II | 19.5 | 19.9 | 19.7 |
| III | 19.7 | 19.5 | 19.8 |
| IV | 19.9 | 19.7 | 20.6 |
| V | 20.1 | 20.1 | 19.6 |
| Size of locality | | | |
| ≤ 2,499 (%) | 46.4 | 43.5 | 48.3 |
| 2,500–14,999 (%) | 10.6 | 13.6 | 10.6 |
| 15,000–99,999 (%) | 8.9 | 9.8 | 10.6 |
| ≥100,000 (%) | 33.9 | 32.9 | 30.3 |

ᵃThe MxFLS was conducted in three rounds: MxFLS-1 in 2002, MxFLS-2 in 2005–2006 and MxFLS-3 in 2009–2012.

## Discussion

We estimate the longitudinal association between daily UPF expenditure per equivalent adult and BMI z-score for age, waist circumference, and waist-to-height ratio in children aged 5–10 years at the baseline round. Using the first tertile of UPF expenditure as the reference category, we find that the second and third tertiles are associated with higher waist circumference

**Table 2. Average household spending on ultra-processed food and beverages, by expenditure tertile.**

| | Round of MxFLS | | |
|---|---|---|---|
| | MxFLS-1 | MxFLS-2 | MxFLS-3 |
| | n = 2,677 | n = 1,474 | n = 1,502 |
| Expenditure on ultra-processed food and beverages[a,b] | | | |
| Lower tertile | 0.41(0.30) | 1.37(2.48) | 1.38(1.55) |
| Middle tertile | 1.67(0.45) | 2.10(2.93) | 2.23(5.73) |
| Upper tertile | 5.17(3.12) | 3.33(3.30) | 3.14(2.43) |

[a]Daily, per adult equivalent (MXN), mean (SD); constant prices as of March 2002.

[b]The sample is classified according to the tertiles of expenditure in MxFLS-1

and waist-to-height ratio in the second and third rounds of the MxFLS. For BMI z-score for age, the results show heterogeneity, emphasizing an interaction between household membership in the top tertile of UPF expenditure, compared to the bottom tertile, and round 2 of the MxFLS.

The increase in the anthropometric indicators studied, associated with greater expenditure on UPF, coincides with a period of accelerated increase in the prevalence of weight excess and obesity in schoolchildren in Mexico, which currently has one of the highest prevalence rates in Latin America [39]. The presence of obesity during childhood and adolescence is associated with the development of short- and long-term alterations that diminish the quality of life. Excess body fat can predispose the individual to musculoskeletal problems, respiratory problems, endocrine disorders that increase the risk of developing chronic non-communicable diseases, as well as psychological problems that can lead to eating disorders. In the long term, socio-environmental factors promote the continuation of obesity in adulthood and the presence of chronic diseases that increase the risk of disability or premature death [33,40].

We study food expenditure, as a proxy for consumption. Although both variables may differ from each other by characteristics such as price or wastage, or the distribution of food within the household, the use of expenditure surveys has shown its usefulness in documenting the advance of UPF in diets in Mexico and other countries[14,41–44]. Thanks to the properties of ultra-processed foods, such as cheap ingredients and long shelf life [3], UPF follow more stable price trajectories than perishable foods and are less prone to waste since they have extended expiration dates. On the other hand, the information analyzed in this study allows us to evaluate the longitudinal association. Taking this into consideration, our results are similar to those produced by Costa et al. for a cohort of children in Brazil regarding waist circumference [11]. They found a positive association between the contribution of UPF to energy intake and the observed 4-year change in waist circumference between preschool age (4 years) and school age (8 years). Another longitudinal study in children in the United Kingdom found a positive association between the top quintile of UPF intake and waist circumference of 0.17 cm for each year of follow-up, which is a moderate increase compared with our estimate [45]. A study involving children and adolescents from eight countries found that a higher contribution of UPF to total energy intake was associated with higher energy density, higher intake of free sugars, and lower intake of fiber, suggesting that UPF consumption is associated with obesity in children and adolescents [46]. Moreover, cross-sectional studies conducted in Brazil and the US also found an association between relatively high UPF consumption and various obesity indicators [47,48]. In addition, it should be noted that we adjusted the models for mother´s BMI, which is consistently positively associated with all three outcomes studied.

**Table 3. Longitudinal association of expenditure on ultra-processed foods and beverages and anthropometric indicators, MxFLS[a], Mexico.**

| Coefficient | Random effects model | | | GEE Model | | |
|---|---|---|---|---|---|---|
| | BMI z-score for age | Waist circumference | Waist-to-height ratio | BMI z-score for age | Waist circumference | Waist-to-height ratio |
| **Round of MxFLS (MxFLS-1 as reference)** | | | | | | |
| MxFLS-2 | **0.58**** | **5.79**** | **0.02**** | **0.60**** | **0.08**** | **0.04**** |
| MxFLS-3 | **0.46**** | **8.90*** | 0.00 | **0.54**** | 0.10 | 0.00 |
| **Interactions** | | | | | | |
| Middle tertile # MxFLS-2 | 0.02 | **4.43**** | **0.02**** | 0.02 | **0.06**** | **0.05**** |
| Middle tertile # MxFLS-3 | **−0.19+** | 2.19 | **0.09+** | **−0.17*** | 0.05 | **0.18+** |
| Upper tertile # MxFLS-2 | **0.19**** | **3.85**** | **0.02**** | **0.19**** | **0.05**** | **0.04**** |
| Upper tertile # MxFLS-3 | −0.15 | −7.99 | 0.04 | −0.13 | −0.14 | 0.09 |
| **Age (years)** | −0.01 | **2.08**** | **−0.00**** | −0.02 | **0.04**** | **−0.01**** |
| **Proportion of time reported watching television (%)** | 0.00 | 0.01 | −0.00 | 0.00 | **0.00*** | −0.00 |
| **Sex** | | | | | | |
| Male | −0.03 | 0.42 | 0.00 | −0.02 | 0.01 | 0.01 |
| **Size of household (number of members)** | **−0.04**** | **−0.21**** | 0.00 | **−0.04**** | **−0.00**** | 0.00 |
| **Mother's years of schooling (no education as reference)** | | | | | | |
| 1–6 years | 0.00 | −0.21 | **−0.01*** | 0.04 | −0.00 | **−0.02*** |
| >6 years | 0.07 | 0.57 | **−0.01+** | 0.11 | 0.01 | −0.01 |
| **Mother's BMI (Underweight or normal as reference)** | | | | | | |
| Overweight (%) | **0.21**** | **1.15**** | **0.00*** | **0.22**** | **0.02**** | **0.01*** |
| Obesity (%) | **0.45**** | **2.44**** | **0.01**** | **0.44**** | **0.04**** | **0.01**** |
| **Mother's marital status (single as reference)** | | | | | | |
| Married/cohabiting (%) | 0.01 | −1.17 | 0.00 | −0.04 | −0.02 | 0.00 |
| Divorced/separated (%) | −0.00 | −1.03 | −0.00 | −0.06 | −0.02 | 0.00 |
| Widowed (%) | 0.21 | 0.42 | 0.01 | 0.13 | 0.02 | **0.03+** |
| **Proportion of food expenditure on non-ultraprocessed foods and beverages (%)** | 0.00 | 0.02 | −0.00 | 0.00 | 0.00 | −0.00 |
| **Wealth Index (Quintile I as reference)** | | | | | | |
| II | 0.06 | **1.22**** | 0.00 | 0.05 | **0.02*** | 0.00 |
| III | −0.01 | 0.62 | **−0.01*** | −0.03 | 0.01 | **−0.02*** |
| IV | **0.09+** | **1.74**** | −0.00 | 0.05 | **0.03**** | −0.01 |
| V | 0.05 | **1.32**** | **−0.01+** | −0.02 | **0.02*** | **−0.01*** |
| **Size of locality (≤2,499 inhabitants as [a] reference)** | | | | | | |
| 2,500–14,999 inhabitants | 0.08 | −0.56 | **−0.01+** | 0.10 | −0.01 | **−0.01+** |
| 15,000–99,999 inhabitants | **0.16*** | 0.74 | −0.00 | **0.12+** | 0.01 | −0.01 |
| ≥100,000 inhabitants | 0.06 | 0.09 | −0.00 | **0.12**** | −0.00 | −0.00 |

** p<0.01, * p<0.05, + p<0.1.

[a]The MxFLS was conducted in three rounds: MxFLS-1 in 2002, MxFLS-2 in 2005–2006 and MxFLS-3 in 2009–2012.

Unlike what they found with waist circumference as an outcome, Costa et al. found no association of UPF consumption with waist-to-height ratio or BMI z-score for age [11]. In contrast, we found a strong association with the waist-to-height ratio, and the results also suggest a positive association with BMI z-score for age.

Cunha et al. found no association between UPF consumption and BMI [12], which differs from our findings. However, they did not adjust for household income. While most of the Brazilian studies were done in a low-income cohort [49], more than half of the participants in the Cunha et al. study could be high-income, given that they study in private schools [12]. In middle-income countries such as Brazil and Mexico, [14,50] income is positively associated with UPF consumption, but mother's years of schooling could prevent the development of obesity [51]. In addition, Cunha et al. studied unstandardized BMI [12]. We studied BMI z-score for age, and unlike Cunha et al., we found a significant association between the high expenditure tertile and MxFLS-2. But even in our sensitivity analysis, taking BMI as the outcome without transforming it to z-score, we found statistical significance in the interaction term between high tertile of UPF expenditure and MxFLS-2 in both models (S1 Table). Arguments in favor of considering unstandardized BMI instead of BMI-for-age z-score claim that it may be a more sensitive method to intrapersonal changes and that it offers better insulation to the regression-to-mean phenomenon compared to BMI-for-age z-score, which is important if there is longitudinal follow-up [29].

It is possible that the association with tertiles of UPF expenditure is more consistently significant for waist circumference and waist-to-height ratio compared to BMI z-score for age because BMI does not necessarily distinguish between fat and fat-free mass in children [40]. Moreover, the estimated models show a positive association of socioeconomic status (proxied by quintiles of asset holdings) with waist circumference, but a negative association with waist-to-height ratio, so the typically observed positive association between linear growth and socioeconomic status could partly explain the lower degree of association of UPF expenditure with BMI-for-age z-score.

We limited our study to children with normal birth weight who were not obese at baseline in 2002. Several studies agree that breastfeeding is a protective factor for childhood obesity [52]. However, we could not adjust for breastfeeding because the information available in the survey does not allow us to determine it for a high percentage of the sample.

## Limitations and strengths of the study

The MxFLS does not have sufficient information to quantify total expenditure on UPF, nor does it provide data on total food expenditure in general. However, comparing the MxFLS food or food groups with those of the ENIGH 2020 (a survey specialized in quantifying household income and expenditure in Mexico), it is estimated that MxFLS food represents on average 72% of total food expenditure in Mexican households.

On the other hand, food expenditure in the MxFLS is measured at a household level and it is not possible to determine how it is distributed within the household, so we do not have an accurate indicator of calorie intake at the individual level nor do we have a direct measure of physical activity at the individual level; this is why we transformed expenditure into an adult-equivalent measure that accounts for differences by age and gender, and incorporated the variable of the proportion of time reported watching television with the intention of mitigating these potential biases. Our categorization into tertiles of UPF expenditure could differ if, for example, children in school settings make purchases that are not reported at the household level or consume despite not acquiring UPF. The correlation between household UPF expenditure and consumption in children and adolescents remains to be studied.

With expenditure, the relative contribution of UPF to total intake could be underestimated compared to consumption data [14]. If the plausibility of the association studied requires high UPF intakes, our associations could be further underestimated.

This is one of the few studies that have analyzed longitudinal data to determine the association between UPF expenditure and various anthropometric indicators and has national representativeness.

Several of the previously published studies that analyzed longitudinal data estimated linear regression models, so they may have somewhat missed the longitudinal structure of the data. We estimated two types of models to explore the longitudinal association and tested several specifications, with which we obtained relatively consistent results so that we had better control of age-year-cohort effects and within-subject correlations.

## Conclusion

Our results suggest that the purchases of ultra-processed foods made by households is a potential driver of abdominal obesity in childhood in the short and medium term. Given the trends in the purchase and consumption of ultra-processed products in the world and Mexico, specific public policies are required to protect the right of children and adolescents to adequate food, which helps to keep the contribution of UPF to energy intake limited from an early age.

## Supporting information

**S1 Table. Longitudinal association of expenditure on ultra-processed foods and beverages and BMI, MxFLS[a], Mexico.** ** $p < 0.01$, * $p < 0.05$, + $p < 0.1$. [a] The MxFLS was conducted in three rounds: MxFLS-1 in 2002, MxFLS-2 in 2005–2006, and MxFLS-3 in 2009–2012.
(XLSX)

**S1 Data. Minimal data set.**
(DTA)

## Acknowledgments

The authors acknowledge the specific contribution of data from MxFLS and INEGI for use in this study. The findings of this study and their interpretation are the responsibility of the authors and do not represent the views or interpretations of the institutions or groups that compiled, collected, or provided the data.

## Author contributions

**Conceptualization:** Mauricio Hernández-F, Mishel Unar-Munguía.

**Data curation:** Erika Lozano-Hidalgo, Lidia Sarahi Peña-Ruiz, Graciela Teruel-Belismelis.

**Formal analysis:** Mauricio Hernández-F, Erika Lozano-Hidalgo, Lidia Sarahi Peña-Ruiz.

**Funding acquisition:** Mauricio Hernández-F.

**Investigation:** Mauricio Hernández-F, Sonia Hernández-Cordero, Mishel Unar-Munguía, Wilfrido A. Gómez-Arias, Erika Lozano-Hidalgo, Lidia Sarahi Peña-Ruiz, Graciela Teruel-Belismelis.

**Methodology:** Mauricio Hernández-F, Sonia Hernández-Cordero, Mishel Unar-Munguía, Wilfrido A. Gómez-Arias.

**Software:** Wilfrido A. Gómez-Arias.

**Supervision:** Sonia Hernández-Cordero, Graciela Teruel-Belismelis.

**Validation:** Mauricio Hernández-F, Sonia Hernández-Cordero, Mishel Unar-Munguía, Wilfrido A. Gómez-Arias, Erika Lozano-Hidalgo, Lidia Sarahi Peña-Ruiz, Graciela Teruel-Belismelis.

**Writing – original draft:** Mauricio Hernández-F.

**Writing – review & editing:** Sonia Hernández-Cordero, Mishel Unar-Munguía, Wilfrido A. Gómez-Arias, Erika Lozano-Hidalgo, Lidia Sarahi Peña-Ruiz, Graciela Teruel-Belismelis.

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
