## [Decision Letter · Decision Letter 0]

16 Oct 2024

PONE-D-24-21652Association of expenditure on ultra-processed foods and beverages and anthropometric indicators in Mexican children: a longitudinal studyPLOS ONE

Dear Dr. Teruel-Belismelis,

Thank you for submitting your manuscript to PLOS ONE. After careful consideration, we feel that it has merit but does not fully meet PLOS ONE’s publication criteria as it currently stands. Therefore, we invite you to submit a revised version of the manuscript that addresses the points raised during the review process.

We look forward to receiving your revised manuscript.

Kind regards,

Shaonong Dang, PhD

Academic Editor

PLOS ONE

Journal Requirements:

“The authors are grateful for the financial support for this research they received from the Research Institute for Equitable Development (EQUIDE for its Spanish acronym) at Universidad Iberoamericana, within the framework of its Call for Research Project Funding 2022.”

“NO authors have competing interests”

Reviewers' comments:

Reviewer's Responses to Questions

**Comments to the Author**

1. Is the manuscript technically sound, and do the data support the conclusions?

Reviewer #1: Yes

Reviewer #2: No

2. Has the statistical analysis been performed appropriately and rigorously? 

Reviewer #1: Yes

Reviewer #2: I Don't Know

3. Have the authors made all data underlying the findings in their manuscript fully available?

Reviewer #1: Yes

Reviewer #2: Yes

4. Is the manuscript presented in an intelligible fashion and written in standard English?

Reviewer #1: Yes

Reviewer #2: No

5. Review Comments to the Author

Reviewer #1: The study explored the longitudinal association between expenditure on ultra-processed foods and beverages (UPF) and changes in anthropometric indicators of obesity with a representative sample. The study was well designed and the statistical analyses were applied appropriately. In the discussion section, I suggest including more studies that investigated the relationship between UPF consumption and obesity and other indicators of body adiposity than those that evaluated variables that were not the subject of their research, such as lipid profile. Below are some references that I suggest the authors consult to improve the discussion.

References

Neri, D., et al, Obesity reviews, 2021. 23 Suppl 1, e13387.

Canella DS et al. PLoS One. 2014 Mar 25;9(3):e92752

Silva FM et al. Public Health Nutr. 2018 Aug;21(12):2271-2279

Juul F et al. Br J Nutr. 2018 Jul;120(1):90-100

Monteiro CA et al. Public Health Nutr. 2018 Jan;21(1):18-26

Reviewer #2: Abstract

background of the abstract is too long. Variables must be mentioned in method section. P values must be added to the results. It is better to write the duration instead of round 2.

Introduction

Introduction is too long and more similar to a literature review. As there are lots of articles on children, it is not logical to mention the works on adults in the introduction. potential mechanisms/justifications for the relation of UPFs intake and obesity must be written. It is more suitable to condense the last paragraph in 2 sentences.

Methods

please name the study's covariates. Add a reference for calculating the wealth index. Table of UPFs is not compete and lacks a lot of food items. 2 main determinants of weigh status are energy intake and physical activity, without measuring them the results could not be reliable.

6. PLOS authors have the option to publish the peer review history of their article (what does this mean? ). If published, this will include your full peer review and any attached files.

**Do you want your identity to be public for this peer review?** For information about this choice, including consent withdrawal, please see our Privacy Policy .

Reviewer #1: No

Reviewer #2: No

---

## [Author Response · Author response to Decision Letter 1]

30 Nov 2024

Response to reviewers

Reviewers' comments:

Reviewer's Responses to Questions

Comments to the Author

1. Is the manuscript technically sound, and do the data support the conclusions?

Reviewer #1: Yes

Reviewer #2: No

Response: Thank you for the comment. The conclusions in lines 367-371 have been revised to ensure consistency with the analyzed data and the results obtained. Although the study does not involve experiments, it was applied a rigorous methodology in conducting this observational study, as detailed in (a) Van Belle G, Fisher L, Heagerty P, Lumley T. Longitudinal Data Analysis. In: Biostatistics: A Methodology for the Health Sciences. In Wiley-Interscience; 2004. p. 728–65, (b) Verbeke, G. & Molenberghs, G. Linear mixed models for longitudinal data. (Springer Science & Business Media, 2000), (c) Liang KY, Zeger SL. Longitudinal Data Analysis Using Generalized Linear Models. Biometrika. 1986;73(1):13-22. These references were also included in the original manuscript.

2. Has the statistical analysis been performed appropriately and rigorously?

Reviewer #1: Yes

Reviewer #2: I Don't Know

Response: The statistical analysis was conducted appropriately and rigorously. Descriptive analysis was performed using standard methodology. As referred to in the manuscript, association analysis was carried out using two generally accepted methods (random-effects linear regression models and generalized estimating equation models), their assumptions were verified, details of their specifications were provided, and their results were thoroughly compared.

3. Have the authors made all data underlying the findings in their manuscript fully available?

Reviewer #1: Yes

Reviewer #2: Yes

Response: Indeed, all the information used in the study is publicly accessible, and the sources for consultation have been provided.

4. Is the manuscript presented in an intelligible fashion and written in standard English?

Reviewer #1: Yes

Reviewer #2: No

Response: Thank you for the observation. The original writing of the manuscript in English was supported by a certified translator. However, the document was reviewed again, with special emphasis on the abstract.

5. Review Comments to the Author

Reviewer #1: The study explored the longitudinal association between expenditure on ultra-processed foods and beverages (UPF) and changes in anthropometric indicators of obesity with a representative sample. The study was well designed and the statistical analyses were applied appropriately. In the discussion section, I suggest including more studies that investigated the relationship between UPF consumption and obesity and other indicators of body adiposity than those that evaluated variables that were not the subject of their research, such as lipid profile. Below are some references that I suggest the authors consult to improve the discussion.

References

Neri, D., et al, Obesity reviews, 2021. 23 Suppl 1, e13387.

Canella DS et al. PLoS One. 2014 Mar 25;9(3):e92752

Silva FM et al. Public Health Nutr. 2018 Aug;21(12):2271-2279

Juul F et al. Br J Nutr. 2018 Jul;120(1):90-100

Monteiro CA et al. Public Health Nutr. 2018 Jan;21(1):18-26

Response: We greatly appreciate the references provided by Reviewer 1 to enrich the discussion. We removed the paragraph that mentioned the studies with lipid profile as an outcome (lines 278-285) and we added several of the suggested references throughout the rest of the discussion.

Reviewer #2: Abstract

background of the abstract is too long. Variables must be mentioned in method section. P values must be added to the results. It is better to write the duration instead of round 2.

Response: Thank you for the suggestions. We shortened the background in the abstract, specified the exposure and outcome variables in the methods, added the p-value and the confidence interval in the results, and replaced the mention of rounds with the follow-up period.

Introduction

Introduction is too long and more similar to a literature review. As there are lots of articles on children, it is not logical to mention the works on adults in the introduction. potential mechanisms/justifications for the relation of UPFs intake and obesity must be written. It is more suitable to condense the last paragraph in 2 sentences.

Response: Thank you for the observation. We shortened the introduction, briefly mentioning the studies conducted in adults. We also mentioned the potential mechanisms for the relation between UPF intake and obesity. We condensed the final paragraph as suggested.

Methods

please name the study's covariates. Add a reference for calculating the wealth index. Table of UPFs is not compete and lacks a lot of food items. 2 main determinants of weigh status are energy intake and physical activity, without measuring them the results could not be reliable.

Response: Thank you for your suggestions. The study covariates were named in the "Covariates" subsection in a manner consistent with the names appearing in Table 1, which outlines the characteristics of the study sample, as well as with the rest of the document. A reference for the calculation of the wealth index has been added. In Fig. 2, it has been clarified that the classified foods and beverages are those with reported expenditure in the MxFLS. We acknowledge that energy intake and physical activity are two main determinants of weight status; therefore, we transformed the expenditure into an adult-equivalent measure that accounts for differences by age and gender and incorporated the variable of the proportion of time reported watching television to mitigate these potential biases, respectively. While the limitations already acknowledged the issue related to the use of adult-equivalent per capita expenditure, we added a mention of the lack of a direct measurement of physical activity.

Note: We included in the cover letter the requested clarification regarding the funding received, the funder's role, and the declaration of interests, as follows:

"I must mention that MHF received a grant (ID 0061) from the Research Institute for Equitable Development (EQUIDE for its Spanish acronym) at Universidad Iberoamericana (https://equide.org/), within the framework of its internal Call for Research Project Funding 2022. The funders had no role in study design, data collection and analysis, decision to publish, or preparation of the manuscript.

The authors have declared that no competing interests exist.

"

---

## [Decision Letter · Decision Letter 1]

6 Jan 2025

Association of expenditure on ultra-processed foods and beverages and anthropometric indicators in Mexican children: a longitudinal study

PONE-D-24-21652R1

Dear Dr. Teruel-Belismelis,

We’re pleased to inform you that your manuscript has been judged scientifically suitable for publication and will be formally accepted for publication once it meets all outstanding technical requirements.

Kind regards,

Shaonong Dang, PhD

Academic Editor

PLOS ONE

Additional Editor Comments (optional):

Authors have addressed the comments from all reviewers, and the manuscript has been revised accordingly and improved for publication.

Reviewers' comments:

Reviewer's Responses to Questions

**Comments to the Author**

1. If the authors have adequately addressed your comments raised in a previous round of review and you feel that this manuscript is now acceptable for publication, you may indicate that here to bypass the “Comments to the Author” section, enter your conflict of interest statement in the “Confidential to Editor” section, and submit your "Accept" recommendation.

Reviewer #1: All comments have been addressed

Reviewer #2: (No Response)

2. Is the manuscript technically sound, and do the data support the conclusions?

Reviewer #1: Yes

Reviewer #2: (No Response)

3. Has the statistical analysis been performed appropriately and rigorously? 

Reviewer #1: Yes

Reviewer #2: (No Response)

4. Have the authors made all data underlying the findings in their manuscript fully available?

Reviewer #1: Yes

Reviewer #2: (No Response)

5. Is the manuscript presented in an intelligible fashion and written in standard English?

Reviewer #1: Yes

Reviewer #2: (No Response)

6. Review Comments to the Author

Reviewer #1: (No Response)

Reviewer #2: (No Response)

7. PLOS authors have the option to publish the peer review history of their article (what does this mean? ). If published, this will include your full peer review and any attached files.

**Do you want your identity to be public for this peer review?** For information about this choice, including consent withdrawal, please see our Privacy Policy .

Reviewer #1: No

Reviewer #2: No

---

## [Editor Report · Acceptance letter]

PONE-D-24-21652R1

PLOS ONE

Dear Dr. Teruel-Belismelis,

I'm pleased to inform you that your manuscript has been deemed suitable for publication in PLOS ONE. Congratulations! Your manuscript is now being handed over to our production team.

Kind regards,

on behalf of

Dr. Shaonong Dang

Academic Editor

PLOS ONE